# Intranodal Injection of Immune Activator Demonstrates Antitumor Efficacy in an Adjuvant Approach

**DOI:** 10.3390/vaccines12040355

**Published:** 2024-03-26

**Authors:** Romano Josi, Anete Ogrina, Dominik Rothen, Ina Balke, Arnau Solé Casaramona, Simone de Brot, Mona O. Mohsen

**Affiliations:** 1Department of BioMedical Research, University of Bern, 3008 Bern, Switzerlandmona.mohsen@unibe.ch (M.O.M.); 2Department of Rheumatology and Immunology RIA, University Hospital Bern, 3010 Bern, Switzerland; 3Graduate School for Cellular and Biomedical Sciences (GCB), University of Bern, 3012 Bern, Switzerland; 4Plant Virology Laboratory, Latvian Biomedical Research and Study Centre, LV-1067 Riga, Latviainab@biomed.lu.lv (I.B.); 5COMPATH, Institute of Animal Pathology, University of Bern, 3012 Bern, Switzerland; simone.debrot@unibe.ch; 6Tajarub Research & Development, Doha P.O. Box 12627, Qatar

**Keywords:** virus-like particles, immune-activator, intranodal, mammary carcinoma, adjuvant therapy

## Abstract

The tumor-draining lymph nodes (tdLN) are the initial site of metastases and are the prime site for generating robust antitumor responses. In this study, we explored the efficacy of a universal immune activator (ImmAct) targeted to the tdLN. This approach can be viewed as an attempt to turn a cold, unresponsive tdLN into a hot, responsive site. The adjuvant antitumor efficacy of our novel intranodal injection was evaluated in an aggressive metastatic mammary carcinoma murine model. The cancer cells were inoculated subcutaneously in the lower quadrant of the mouse to provoke the tdLN (inguinal lymph node). The study encompasses a range of methodologies, including in vivo and in vitro assays and high-dimensional flow cytometry analysis. Our findings demonstrated that intranodal administration of ImmAct following the dissection of the primary tumor led to improved tumor-free survival and minimized weight loss. ImmAct led to both local and systemic alterations in the cellular and humoral immunity. Additionally, after ImmAct treatment, non-responders showed a higher rate of exhausted CD8^+^ T cells compared to responders. Indeed, our innovative approach surpassed the gold standard surgery of sentinel lymph node excision. Overall, intranodal administration of ImmAct yielded a robust antitumor immune response, offering protection against micrometastases and relapse.

## 1. Introduction

The formation of distant metastases and the dissemination of cancer cells are thought to be the main causes of cancer-related mortality. The metastatic cascade typically initiates in the tumor-draining lymph nodes (tdLNs), also known as the sentinel lymph nodes (SLNs), where locoregional metastases take root [1,2]. From the tdLNs, tumor cells can disseminate to distant LNs and organs in the body [3]. Tumor lymphangiogenesis, involving the proliferation of lymphatic vessels, supports the migration of tumor cells to tdLNs and may be associated with an unfavorable prognosis [2,4]. Generally, lymphatic vessels play a crucial role in transporting immune cells and tumor antigens (tAgs) to tdLNs, where both local and systemic immunity are coordinated through innate and specific immune mechanisms. The tdLN serves as the initial site where naïve T cells are primed against tAgs. To evade the induced immune response, the tumor must effectively suppress these primed T cells. Given the high density of dendritic cells (DCs) and T cells in the LNs, costimulatory signals for T and B cells can be provided, often in a bystander manner [5]. As the tumor progresses, changes occur in the immune composition of the tdLN. For instance, an altered CD4/CD8 T cells ratio and an increase in T regulatory cells (Tregs) become more pronounced, especially following metastatic involvement in melanoma, breast cancer, and cervical cancer [6,7,8]. As metastasis in the tdLN expands, there is a concurrent increase in the recruitment of myeloid-derived suppressor cells (MDSCs), accompanied by the release of immunosuppressive exosomes and soluble mediators [9]. A pre-metastatic niche may arise as a result of several of these alterations occurring even before the appearance of tumor cells [10].

Past efforts in therapeutic cancer vaccines primarily targeted tumor-associated antigens (TAAs), yielding modest immune responses due to immune tolerance towards self-antigens. Therefore, neoantigens, not expected to induce T cell tolerance, have attracted attention. Personalized vaccines directed against individual tumor neoantigens, using long peptides or RNA, have shown promising outcomes in mice [11] and humans [12,13]. However, challenges in this approach include identifying individual antigens, rendering epitopes immunogenic in humans, and the substantial tissue required for characterizing the immunopeptidome [14]. 

The fact that tumors and their microenvironment offer relatively little support for T cell activation may be the reason for intratumoral immunotherapy comparatively modest level of success as of yet. This is fundamentally different in the LNs, as they are highly specialized structures to effectively activate T cells. Together with the presence of metastatic tumor cells (providing tAgs), LNs represent much better conditions to promote antitumor T cell response than solid tumors. Therefore, intranodal treatments that reverse immune suppression and promote T cell activation hold great promise for harnessing the patient’s immune system for efficient fighting against cancer. 

The concept of intralymphatic/intranodal vaccination was initially investigated in the 1970s. Juillard et al. utilized this approach to augment the effectiveness of tumor cell-based vaccination in dogs [15,16,17]. Subsequently, in the 1980s, researchers found that injecting nanograms of proteins via intranodal delivery was enough to elicit a humoral immune response [18]. A biodistribution study in mice revealed that intranodal injection of antigens resulted in a concentration in the dLNs that was 100-fold higher compared to subcutaneous (s.c.) injection [16]. Human studies have demonstrated that twenty minutes post-injection into an inguinal lymph node (iLN), the protein had migrated to deeper pelvic LNs, a phenomenon not observed after s.c. injection [16]. Several pre-clinical and clinical studies have centered on the intralymphatic administration of drugs, immunotherapies, or vaccines. In humans, drugs are typically introduced into an iLN under ultrasound guidance, a routine procedure performed by experienced radiologists [19]. In the context of cancer immunotherapy, DCs pulsed with antigens and administered intranodally in melanoma patients have shown a propensity to localize in the paracortex of LNs [20,21]. While some clinical trials employing intranodal DC therapy have suggested enhanced immune responses, others have failed to demonstrate a significant advantage of the intranodal route compared to intradermal injection [22,23,24]. 

Virus-like particles (VLPs) are nanoparticles structured in a repetitive icosahedral geometry. Although VLPs resemble their parent viruses, they lack the genetic material necessary for reproduction which contributes to their good safety profile [25]. A pathogen-associated structural pattern (PASP) that can trigger potent innate and adaptive immune responses is represented by the surface geometry of VLP. In addition to being used and sold as part of preventative vaccinations against the hepatitis B virus (HBV) and human papillomavirus (HPV), VLPs are currently being extensively studied in the area of cancer immunotherapy [26]. Among others, we have successfully designed and developed different therapeutic cancer vaccines utilizing VLPs and have provided several proofs of concepts in mice [11,27,28,29]. Recently, we have focused on local delivery of VLPs as effective immunotherapy [29]. We formulated our plant-derived VLPs (naked VLPs, i.e., do not display any tAgs) with a depot-forming microcrystalline tyrosine adjuvant (MCT) for intratumoral administration in mice harboring the aggressive B16F10 melanoma [29]. The therapy activated the immune system through multiple immunological pathways resulting in considerable antitumor effects. Importantly, this strategy does not require the knowledge of tAgs, meaning that the same therapy can be used for different cancer types, thus facilitating translation to humans. 

In this study, we evaluated the antitumor efficacy of our plant-derived VLP, henceforth referred to as immune activator (ImmAct) for intranodal administration, directly in the tdLN, aiming to protect mice against local and systemic relapse. We employed an adjuvant approach using the aggressive metastatic murine mammary carcinoma model 4T1. 

Our study provides a proof of concept of an effective and novel vaccination strategy with naked VLPs that can readily be implemented for clinical use. 

## 2. Results

### 2.1. Establishing a Murine Tumor Model with Metastatic Lymph Nodes

In this study, we utilized our immunogenic plant-virus-like nanoparticles derived from cowpea chlorotic mottle VLPs (CCMV_TT_-VLPs referred to as ImmAct) [26]. We produced and fully characterized the VLPs, as described in Figure 1A–D, and the 45 nm icosahedral nanoparticles package ssRNA-TLR7/8 ligand during the expression process in the *E. coli* host system. CCMV_TT_-VLPs also incorporate an internal tetanus-toxin epitope (TT), known to activate T helper (T_H_) cells [26]. We then worked on establishing a murine model in which tumor cells from the established primary tumor are capable of metastasizing to the tdLNs. We used the aggressive 4T1 murine mammary carcinoma cells which have the potential to form lymphangiogenesis and metastatic disease. Briefly, 10^5^ cells were injected s.c. in the lower quadrant of the mouse facilitating tumor drainage to the inguinal LN, equivalent to tdLN in this study (Figure 1E). We confirmed the metastatic disease in the tdLN 11 days post-tumor-cell inoculation by performing an in vivo experiment with labeled nanoparticles, quantitative real-time PCR (qrtPCR), and histological analyses. For the in vivo experiment, we labeled CCMV_TT_ with AF488 and injected 15 μg directly into an established primary tumor. Twenty-four hours later, we collected the tdLN and non-tdLN (ntdLN) to assess the presence of the labeled VLPs. Our data show that significantly more CD11b^+^ cells picked up the labeled VLPs in the tdLNs; however, no uptake of the labeled VLPs was detected in the non-tdLNs (*p*. 0.0004) (Figure 1F,G). We performed hematoxylin and eosin (HE) staining of the tdLNs and confirmed the spread of 4T1 tumor cells into the tdLN (Figure 1H). Finally, we conducted qrtPCR to assess the expression levels of four distinct neoantigens identified previously by us in the 4T1 cell line within the tdLNs. The findings demonstrated comparable expression levels of the selected neoantigens in the tdLNs and 4T1 tumors (Figure 1I–L). Collectively, these data affirm the appropriateness of our murine model for evaluating the immune-activating properties of our VLPs using the intranodal administration strategy.

### 2.2. Intranodal Injection of ImmAct Demonstrates Antitumor Efficacy in an Adjuvant Setting

The icosahedral CCMV_TT-_VLPs were used as naked VLPs (i.e., they do not display any cancer-specific antigens on their surface); hence, they were used in this study as an immune activator, referred to hereafter as ImmAct. We assessed the antitumor efficacy of an intranodal injection of ImmAct in an adjuvant setting (i.e., after surgical removal of the primary tumor). Briefly, 10^5^ 4T1 cancer cells were injected s.c. into the flank of mice as established in result 1 to provoke tumor metastases to the tdLN. Eleven days later, the primary tumor was dissected under anesthesia, and 30 μg of ImmAct was administered intranodally in the tdLN, followed by a booster dose on day 18 (Figure 2A,B). In parallel, a control (CTRL) group of mice was injected with PBS only. Over 60 days, the mice’s general health, weight loss, and local recurrence were observed. Here, the term “local relapse” describes the tumor growth back at the initial site of the tumor cell inoculation. Mice were euthanized when showing a local relapse of a tumor reaching the ethically maximal size of 1 cm^3^ and/or if the mice had lost 10% of their original body weight. Primary dissected tumors in both groups—before intranodal adjuvant treatment—showed no statistical differences as expected (*p*. 0.8943) (Figure 2C). Mice treated intranodally with ImmAct showed a significantly lower weight loss in comparison to the control group (*p* < 0.0001) (Figure 2D). Our results reveal 60% tumor-free survival in the group that received ImmAct intranodally compared to the PBS CTRL (*p*. 0.0009) (Figure 2E); here, tumor-free survival describes the period during which mice show no sign of local relapse or weight loss due to metastases. To assess the histological changes in the tdLN following intranodal treatment with ImmAct, we performed HE staining. The results reveal overall enlargement of the tdLN with increased lymphocyte numbers in the cortical region and formation of secondary follicles, as well as numerous plasma cells and macrophages in the medulla, indicative of reactive hyperplasia (Figure 2F). These results confirm the antitumor efficacy of intranodal immunotherapy with ImmAct. 

### 2.3. Intranodal Administration of ImmAct Results in Local and Systemic Changes in Cellular and Humoral Immunity

To investigate both the local and systemic effects of our novel intranodal immunotherapy using ImmAct, we gathered blood samples on day 22, corresponding to the prime and boost phases, as well as spleen and treated tdLN specimens obtained upon reaching the humane endpoint in mice. When analyzing the blood, no notable distinctions were observed in the proportion of CD8^+^ T cells or MDSCs between the two groups (Figure 3A panel). However, a significant rise in CD4^+^ T cells was identified following intranodal administration of ImmAct (*p* < 0.0001), accompanied by a substantial reduction in B cells marked by the CD19 surface marker (*p*. 0.0008) (Figure 3A). Analysis of the spleen revealed no changes in the percentage of CD8^+^ or CD4^+^ T cells; however, a significant increase in B cells percentage was reported (*p*. 0.0003), accompanied by a significant decrease in MDSCs characterized by CD1b^+^Ly6G^+^ and CD11b^+^Ly6C^+^ (*p*. 0.0002 and *p* < 0.0001 respectively) (Figure 3B panel). Evaluation of the tdLNs yielded comparable findings to the blood, with a significant increase in CD4^+^ T cells (*p*. 0.0437) in mice receiving intranodal ImmAct, while the percentage of B cells was lower (*p*. 0.00154) (Figure 3C panel). Subsequently, we categorized mice treated intranodally with ImmAct into responder and non-responder cohorts. Responders are referred to mice that showed tumor-free survival and no local recurrence during the experimental course. We conducted an in-depth analysis using unbiased and high-dimensional flow cytometry data (concatenating four samples from each group). Focusing on the CD8 surface marker and exhaustion markers PD-1 and TIM-3, we applied the non-linear dimensionality reduction technique, t-stochastic neighbor embedding (tSNE), followed by FlowSOM analysis to identify distinct T cell clusters (Figure 3D). Five distinct T cell clusters (Pop 0-4) were identified, as illustrated in the generated heatmaps (Figure 3E). Notably, Population 0 increased in the non-responder group, while Population 1 decreased. Both of these CD8 populations showed no expression of PD-1 or TIM-3 (Figure 3E), emphasizing the need for additional markers to investigate these populations thoroughly. Certainly, population 2 (expressing TIM-3) and populations 3 and 4 (expressing PD-1) exhibited an increase in the non-responder group (Figure 3E,F). Collectively, these data suggest that intranodal administration of ImmAct results in local and systemic changes in cellular and humoral immunity. 

### 2.4. Intranodal Treatment with ImmAct Outperforms Sentinel Lymph Node Excision

Subsequently, we compared the efficacy (specifically, tumor-free survival) of the standard SLNE surgery with our intranodal immunotherapy with ImmAct (Figure 4A). Three groups were included: CTRL, intranodal treatment with ImmAct, and standard SLNE surgery. The treatment regimen with ImmAct was described in Figure 2A. For group 3, SLNE was performed on day 11 combined with the dissection of the primary tumor. No differences in the primary tumor size was detected among the three groups, as expected (Figure 4B). The mice were monitored for local relapse, metastases, and general health over a period of 60 days. Metastatic disease was confirmed upon lung tissue examination during necropsy for metastatic nodules. On day 22, the blood analysis revealed consistent results with our previous findings. Intranodal immunotherapy with ImmAct did not alter the percentage of CD8^+^ T cells, but led to a significant increase in CD4^+^ T cells compared to the CTRL group or the group that underwent SLNE (*p* < 0.0001 and *p*. 0.0221 respectively) (Figure 4C panel). Additionally, the percentage of CD19^+^ B cells significantly decreased in the ImmAct-immunized group compared to the CTRL or the SLNE group (*p*. 0.0001 and *p*. 0.0403 respectively) (Figure 4C panel). A modest reduction in MDCS, particularly in the CD11b^+^ Ly6G^+^ cells, was observed in the group treated with ImmAct compared to the CTRL group (*p*. 0.0314) (Figure 4C panel). Intranodal injection of ImmAct significantly prolonged the tumor-free survival rate when compared to the other groups (Figure 4D). Specifically, 70% of mice in the CTRL group had a local relapse and metastases, ~28% showed only local relapse, and ~2% experienced tumor-free survival (Figure 4E). Mice that underwent SLNE only experienced 15% tumor-free survival, 35% local relapse, and 50% local relapse combined with lung metastases. The best results were obtained with intranodal ImmAct immunotherapy, resulting in 70% tumor-free survival, 15% local relapse, and 15% local relapse combined with metastases (Figure 4E). Figure 4F shows representative images of lungs of mice treated with ImmAct versus SLNE.

### 2.5. Essential Roles of Cellular and Humoral Immune Components in Orchestrating the Efficacy of ImmAct

To investigate the contribution of cellular and humoral immune cells to the observed antitumor efficacy of ImmAct, we selectively depleted CD8^+^ and CD4^+^ T cells, as well as CD19^+^ B cells. Five distinct mice groups were included in the analysis: a CTRL group, receiving intranodal PBS treatment, a group treated intranodally with ImmAct (following the regimen outlined in Figure 2A), and three additional groups subjected to the same intranodal ImmAct treatment along with intravenous administration of anti-CD8 monoclonal antibodies (mAbs), anti-CD4 mAbs, or anti-CD19 mAbs (Figure 5A). The schedule and dose of the mAbs was carried out according to the described method section. Throughout a 60 day observation period, mice were carefully monitored for local relapse, metastases, and overall health. Metastatic disease was confirmed through lung tissue examination during necropsy, revealing the presence of metastatic nodules. Flow cytometry analysis conducted 48 h post mAbs administration confirmed efficient T and B cell depletion, reaching approximately 100% depletion efficiency (Figure 5B–D). Primary dissected tumors in both groups—before intranodal adjuvant treatment—showed no statistical differences, as expected (*p*. 0.4165) (Figure 5E). Consistent with our earlier findings, intranodal administration of ImmAct resulted in a tumor-free survival rate of 60%. Notably, the immunotherapeutic efficacy of ImmAct was significantly diminished in both T-cell- and B-cell-depleted groups. As previously demonstrated, the CTRL group exhibited poor survival (Figure 5F). In the subsequent analysis, we evaluated the percentage of local relapse, local relapse combined with lung metastases, and tumor-free survival in each experimental group (Figure 5G). Results revealed a 70% occurrence of local relapse combined with lung metastases in both the CTRL group and the group treated with ImmAct while depleting CD8^+^ T cells. Depletion of CD4^+^ T cells or B cells during ImmAct treatment resulted in a 70% local relapse rate and a 30% rate of local relapse with metastases. Once again, intranodal treatment with ImmAct demonstrated a consistent 70% tumor-free survival, accompanied by a 15% rate of local relapse or local relapse with lung metastases. Depletion of CD8^+^ T cells showed an outcome similar to the CTRL group, underscoring the crucial role of CD8^+^ T cells in the induced antitumor efficacy of intranodal ImmAct treatment. Depleting CD4^+^ T cells or B cells demonstrated fewer overall lung metastases compared to the depletion of CD8^+^ T cells, but it led to an increase in the percentage of local relapse. These data suggest an essential role played by these two immune components, a topic currently under investigation in our research.

## 3. Discussion

In this study, we explored the adjuvant antitumor efficacy of plant-derived nanoparticles designed for intranodal injection directly into the tdLN using an aggressive metastatic mammary carcinoma murine model. Our strategy mirrors the in situ/intratumor immunization strategy [29], aiming to convert immunosuppressed tdLNs into active, hot ones. 

Our results reveal a significant antitumor response evidenced by protection against local and systemic tumor relapse and minimized weight loss. The intranodal administration of ImmAct led to both local and systemic alterations in cellular and humoral immunity, primarily characterized by an enhancement of CD4^+^ T cells in the blood and the tdLN in the group treated with ImmAct. Furthermore, differentiating between responder and non-responder mice after treatment with ImmAct revealed an increased percentage of CD8^+^ T cells in the tdLN expressing exhaustion markers such as PD-1 and TIM-3. Indeed, our innovative approach surpassed the gold standard surgery of SLNE. 

Our immunotherapy with ImmAct was directly administered into a metastatic tdLN to induce a robust local and systemic immune response and to enhance tumor-free survival. In humans and dogs, intranodal injection of inactivated tumor cells has been tested with promising results [17,30,31]. Additionally, the administration of MHC-I binding peptides into LNs or the spleen has induced robust CD8^+^ T cell response, protecting from tumor growth or viral replication in mice [32]. Furthermore, intranodal immunization has significantly augmented the effects of plasmid DNA [33,34] and RNA vaccination in mice [35,36]. Intranodal immunotherapy utilizing our ImmAct potentially offers significant benefits compared to other intranodal approaches. The distinct advantage of our method lies in the use of VLPs, which are not only safe, but also equipped with a potent TLR-ligand, coupled with a tetanus toxin (TT) internal epitope. This unique combination is designed to amplify the immune response in future translation into humans, leveraging the widespread immunity to tetanus already established in most individuals. This aspect is particularly advantageous in enhancing the immune responses of elderly patients or those with cancer, who often experience compromised immune functionality. 

The advantage of intranodal immunization lies in the high efficiency of reaching both naïve T cells and tumor-infiltrating lymphocytes (TILs) in these LNs, which, have been suppressed by the tumor [37]. It is well-established that VLPs have an inherent capacity to provide additional T cell help beyond the primary benefits of the VLP itself, as they induce VLP-specific CD4^+^ T cells [38]. This phenomenon likely accounts for the significant increase in CD4^+^ T cell response observed in both the bloodstream and the tdLNs within the ImmAct-treated group. Intriguingly, this augmentation in CD4^+^ T cells was accompanied by a noteworthy reduction in B cells at these two locations, contrasted by an increase in the spleen. These findings hint at the possibility that activated B cells may be migrating toward and accumulating in the spleen after their activation. This redistribution of B cells may also suggest that ImmAct is altering the homing signals or the environment within the tdLN, making the spleen a more favorable site for B cell activation. 

Although there is no discernible rise in CD8^+^ T cells observed in the blood, spleen, or tdLN following ImmAct immunization, depletion of this population nullified the antitumor effect, leading to heightened local and distant relapse of approximately 70%. The anti-tumor efficacy of ImmAct was found to be critically dependent on CD8^+^ T cells, even though it was partially dependent on CD4^+^ T cells and B cells. This was demonstrated by the significantly reduced protection observed upon in vivo depletion of CD8^+^ T cells, supporting our previous findings highlighting the critical role of CD8^+^ T cells in the s.c. or intratumoral administration of VLP-based vaccines or immune enhancers [27,29]. Exploring combination therapy with checkpoint inhibitors may be a strategy to further enhance the efficacy of ImmAct, potentially with synergistic effects for improving immune responses and outcomes in cancer treatment.

Our data demonstrate a significant improvement in tumor-free survival among mice treated with ImmAct compared to the CTRL group. Remarkably, the intranodal administration of ImmAct exhibited superior efficacy over the widely accepted standard medical procedure of SLNE surgery, showing a 70% tumor-free survival rate in contrast to 15% achieved by SLNE alone. This outcome is particularly noteworthy as SLNE has long stood as the gold standard of therapy for preventing metastatic disease in cancer patients. In stage III melanoma patients, treatment has dramatically improved by de-escalating surgery of occult metastases in regional LNs and the introduction of modern immunotherapies [12]. Recently, Trophy et. al., compared the outcome of adjuvant therapy versus observation in melanoma patients with a positive SLN that did not undergo complete LN dissection [13]. The study found a significantly better 24 month metastases-free survival in patients undergoing adjuvant therapy (86% vs. 59%), highlighting the importance of this approach [13]. 

Remarkably, the analysis of responder and non-responder cohorts post-ImmAct immunotherapy uncovered intriguing insights into CD8 populations. Specifically, the non-responder group exhibited a distinctive rise in exhaustion markers, anti-PD1 and TIM-3. In recent research, findings have indicated that TIM-3 is a component of a complex module encompassing several co-inhibitory receptors, commonly known as checkpoint receptors. These receptors tend to be concurrently expressed and regulated on T cells experiencing dysfunction or exhaustion in the context of chronic viral infections and cancer [39]. These data may suggest that a higher proportion of non-responders experience T cell dysfunction which could be due to prolonged antigen exposure [40]. Further exploration of these findings holds promise for optimizing the therapeutic dose of ImmAct to overcome exhaustion and enhance the immunotherapy outcome. We are presently investigating the inclusion of additional T cell exhaustion markers, aiming to enhance the categorization of T cell states. 

Adopting an adjuvant approach could pose challenges for future human applications, especially since most patients undergo SLNE during the surgical removal of solid tumors. Consequently, evaluating our ImmAct in a neoadjuvant immunotherapeutic context presents an intriguing avenue and is currently underway in our laboratory.

## 4. Conclusions

To sum up, our work effectively illustrates the efficacy of intranodal immunotherapy that utilizes the advantage of our novel immune activator derived from plant nanoparticles that are loaded with potent immunological stimuli. The administration of ImmAct in an adjuvant setting (i.e., post-primary tumor dissection) involving only a prime and a boost injection not only highlights the robust efficacy of our immunotherapy, but also paves the way for future optimization in preparation for human translation.

## 5. Methods

### 5.1. CCMV_TT_-VLPs Cloning, Expression, and Production

The coat protein sequence of the CCMV VLPs can be obtained from the GenBank (accession no. AAA46373). The sequence, cloning, expression, and production of our plant-derived VLPs are detailed in [26]. In short, a cloned copy of the CCMV coat protein gene was used in PCR mutagenesis to insert the tetanus toxin coding sequence, resulting in the cloning of CCMVtt. *E. coli* C2566 cells were used for cloning and plasmid amplification. Isopropyl-β-D-thiogalactopyranoside (IPTG) was used to initiate the expression, and the pellet was collected using low-speed centrifugation following the incubation period. Using ultracentrifugation in a sucrose gradient ranging from 20% to 60% sucrose, VLPs were isolated from cellular proteins. The fractions were loaded in an SDS-PAGE gel. To eliminate the sucrose, fractions containing CP proteins were mixed and dialyzed. We used an improved version in this work, which is currently undergoing patenting. 

### 5.2. Dynamic Light Scattering Measurement (DLS)

A high-precision micro quartz cuvette (ZEN2112, HellmaAnalytics, sample volume 12 μL) was used to analyze purified, undiluted VLPs at a concentration of 1 mg/mL using Zetasizer Nano ZS equipment (Malvern Instruments Ltd., Malvern, UK). We calculated the average hydrodynamic diameter of VLPs using three consecutive measurements. The Zetasizer software (version 8.01, Malvern Instruments Ltd., Malvern, UK) was used to analyze the results.

### 5.3. Electron Microscopy

The visualization of purified VLPs involved uranyl acetate negative staining. Initially, 5 µL of the sample at a concentration of 1 mg/mL was adsorbed onto carbon formvar-coated 300 Mesh Copper grids (Agar Scientific, Stansted, UK), with the preparation of 2 grids per sample. After a 3 min incubation, the grids underwent washing with 1 mM ethylenediaminetetraacetic acid (EDTA) and subsequent negative staining using a 0.5% uranyl acetate aqueous solution. Analysis was carried out using a JEM-1230 electron microscope (JEOL, Tokyo, Japan) operating at an accelerating voltage of 100 kV. A minimum of five micrograph pictures were captured for each sample.

### 5.4. SDS-Page and Gel Electrophoresis 

For the evaluation of CCMV_TT_-VLPs, 10 μg of the sample was mixed with 4x Laemmli buffer (comprising 100 mM TRIS-HCl pH 7.0, 4% SDS, 5% β-mercaptoethanol (β-ME), 50% glycerine, 0.4% bromphenol blue) and heated at 95 °C for 10 min. Subsequently, this prepared mixture was loaded onto a self-made 12.5% SDS-PAGE gel, following the protocol outlined in [41], and run for 1 h at 250 V, 20 mA. To eliminate SDS, the gel underwent a 10 min wash with a fixing solution of 10% ethanol and 10% acetic acid, followed by staining with Coomassie G250 dye (composed of 20% (*v*/*v*) ethanol, 2% (*w*/*v*) trichloroacetic acid, and 0.5 g/L Coomassie G250) overnight. Page Ruler™ Prestained Plus Protein Ladder (cat. no. 26619, Thermo Fisher Scientific, Waltham, MA, USA) was used as a reference marker. For the gel electrophoresis, 10 μg of the sample was loaded onto a 0.8% agarose gel supplemented with 0.2 μg/mL Ethidium Bromide. A Master Mix DNA ladder (cat. no. SM0331, Thermo Fisher Scientific, Zürich, Switzerland) was used as a reference.

### 5.5. Labelling CCMV_TT_-VLPs

CCMV_TT_-VLPs were labeled with Alexa Fluor 488 as per the manufacturer’s instructions (Thermo Fischer Scientific, Zürich, Switzerland, A10235) and stored at −20 °C. Briefly, 100 μg of CCMV_TT_-VLPs was mixed with 1 μL of AF488 in an Eppendorf tube (protected from light) on the shaker for 1 h at 400 rpm. Zeba^TM^ spin columns (7k MWCO) were prepared as per the manufacturer’s instruction (Thermo Scientific, Zürich, Switzerland). CCMV_TT-_AF488 were transferred to Zeba columns and spun for 2 min at 1000× *g*. 

### 5.6. Cell Culture

Cells (4T1) were grown in Falcon 75 cm^2^ Flask (Corning, NY, USA, 353136,) with DMEM (GIBCO) medium supplemented with 10% fetal bovine serum (FBS) (Thermo Fisher Scientific, Zürich, Switzerland 16250086), 1% penicillin-streptomycin (p/s) (Merk, Darmstadt, Germany, I9657). Cells were rinsed three times with 1× PBS to eliminate all cell culture medium when they reached 80% confluency. A total of 0.5% Trypsin-EDTA (Thermo Fisher Scientific, Zürich, Switzerland, 15400-054) was added and the flask was incubated for 7 min at 37 °C. Cells were centrifuged at 300× *g* for 5 min. Cells were resuspended in a culturing medium and kept on ice until further use. The cells were counted using Cellometer mini (Nexcelon Bioscience, Lawrence, MA, USA) and then resuspended in the appropriate amount of medium to inject 10^5^ cells/mouse. Using the Microsart AMP Mycoplasma Kit (Sartorius, Göttigen, Germany, SMB95-1001), mycoplasma contamination was ruled out.

### 5.7. Mice

All in vivo experiments utilized wild-type female Balb/cOlaHsd mice (8–12 weeks) and were purchased from Harlan. The mice were housed and bred in the pathogen-free animal facility at the University of Bern. All animal procedures were conducted under license BE43/21, adhering to the Swiss Animal Act (455.109.1—5 September 2008).

### 5.8. In Vivo Drainage of VLPs to tdLN

Under isoflurane anesthesia (Attane, 800-544-7521), 10^5^ 4T1 cells were s.c. inoculated in the lower left quadrant of wild-type BALB/cOlaHsd mice. Every two days, mice were monitored to measure the progress of tumors and their overall health. Eleven days post-tumor inoculation, 15 μg of CCMV_TT_-AF488 was injected directly into the primary palpable tumor under isoflurane anesthesia. Twenty hours later, the tdLN and the collateral non-tdLN were collected and kept on ice. The lymph nodes were treated with collagenase D (Roche, Basel, Switzerland, 11088858001) and DNase I (Roche, 10104159001) in a DMEM medium for 30 min at 37 °C. The ACK buffer (Sigma-Aldrich, Burlington, MA, USA, R7757) was used to lyse red blood cells (RBCs) after the lymph nodes had been smashed through a 70 μm cell strainer. Cells were transferred to a 96 V-shape well plate and stained with Fc block CD16/CD32 (FRC/4G8) (BD Bioscience, Franklin Lake, NJ, USA, 553142) for 5 min at RT in the dark. The plate was centrifuged at 1200 rpm for 5 min and flicked. Cells were then stained with 7-AAD viability staining solution (Thermo Fisher Scientific, 00-6993-50) 1/5000 and anti-mouse CD11b PE-Cy7 clone M1/70 (Thermo Fisher Scientific, 25-0112-82) for 10 min at RT in the dark. The plate was centrifuged at 1200 rpm for 5 min and flicked. After surface staining, cells were washed twice and resuspended in 400 μL FACS buffer. Sample analysis was performed using a BD LSRII and evaluated with Flojo (V.10) and GraphPad Prism (V10). The gating strategy involved a sequential selection of singlets, live/dead followed by monocytes and further categorized into CD11b^+^ AF488^+^ cells. 

### 5.9. Real-Time Quantitative PCR

We assessed four different neoantigens in the tdLNs and ntLNs in mice harboring 4T1 cells. A total of 10^5^ 4T1 cells were s.c. inoculated in the lower left quadrant of wild-type BALB/cOlaHsd mice. The tdLN and the ntdLN were dissected on day 11, stored in RNA (Merk, Darmstadt, Germany), and kept at −20 °C. For RNA extraction, we followed the manufacturer’s instructions (NucleoSpin RNA Kit) and the RNA content was measured by Nanodrop. The cDNA was produced using a High-Capacity RNA-to-cDNA Kit as per the manufacturer’s instructions and the cDNA content was measured with Nanodrop. To amplify the mutated region, three primer pairs of about 24 bp each were created, resulting in three amplicons of about 200 bp each. After the PCR products were electrophoresed and gel-extracted, the Zymoclean Gel DNA Recovery Kit was used to purify the amplicons that were predicted in size. Following PCR amplification, amplicons were Sanger sequenced using the primers. The housekeeping gene was included.

### 5.10. Intranodal Administration of ImmAct

Under isoflurane anesthesia (Attane, 800-544-7521), 10^5^ 4T1 cells were s.c. inoculated in the lower left quadrant of wild-type BALB/cOlaHsd mice. Mice were followed every 2 days to assess tumor growth and general health score. Eleven days later, the primary tumor was surgically dissected as follows: the weight of the mice was recorded, and 50 μL/25 g of narcotic mixture was injected s.c. The narcotic mixture consisted of Dormitor (Medetomidine Hydrochloride) at a concentration of 1 mg/mL and a dosage of 0.4 mg/kg, Dormicum (Midazolam) at a concentration of 5 mg/mL and a dosage of 4 mg/kg, and Fentanyl at a concentration of 0.05 mg/mL and a dosage of 0.04 mg/kg. Ophthalmic ointment (Paralube^®^) was applied to the mouse’s eyes to avoid the drying of the cornea. Ten minutes later, the mice were assessed for reflex responses, confirming their state of full sedation. Mice were positioned on their backs to expose the iLN, the skin of this area was shaved using a trimmer, and disinfection was carried out with Betadine. The hip joint was bent to an approximately 90° angle and a small incision of <5 mm through the skin was made using sterile curved micro-dissecting forceps and a surgical scissor. The incision was widened using the tip of the surgical scissor which enlarged the incision to about 10 mm. The iLN was localized and immobilized using the curved forceps and the surgical scissor. The iLN appeared greyish within the surrounding fat tissue. Using an insulin syringe, 0.5 mL, 30 μg, and 10 μL of the naked CCMVTT-VLPs (ImmAct) or Phosphate buffer reagent (PBS) was directly injected into the iLN. If the lymph node swelled, the injection was deemed successful. The incision was sealed using sterile 9 mm wound clips inserted into a 9 mm auto-clip applier. Clips were removed after 10–12 days. The mice were boosted with the same method after one week. Mice were monitored closely as per our license BE43/2021. 

### 5.11. Histology

The collected iLNs were fixed with 4% buffered formalin (Formafix GmbH, Zürich, Switzerland) for at least 24 h and routinely processed and stained with Hematoxylin and Eosin (HE) for histological examination. The iLN tissue sections were evaluated for the presence of neoplastic cells and any other histopathological changes by a board-certified veterinary pathologist (SdB). 

### 5.12. Blood, Spleen, and Lymph Node Processing

On day 22, blood from the tail vein of mice was collected in Eppendorf tubes containing 0.5 M EDTA in 1× PBS. For all the steps, centrifugation for 5 min at 1200 rpm was used. Blood was centrifuged and the supernatant was aspirated. Red blood cells (RBCs) were lysed using AmmoniumChloride—Potassium (ACK) buffer on ice for 5 min. Cells were centrifuged for 5 min at 1200 rpm and washed twice with FACS buffer (1× PBS containing 0.1% Bovine Serum Albumin (BSA)). After the final wash, the cells were suspended in FACS buffer, transferred to a 96 U-shape well plate, and kept on ice for staining. The spleens were collected on ice and smashed using 70 μm cell strainers, and RBCs were lysed with ACK buffer. Cells were centrifuged and washed as summarized above. A total of 10^6^ splenocytes were transferred to a 96 U-shape well plate and kept on ice for staining. For lymph node processing, tdLNs were collected on ice and then treated with collagenase D (Roche, 11088858001) and DNase I (Roche, 10104159001) in a DMEM medium for 30 min at 37 °C. The lymph nodes were smashed using 70 μm cell strainers, and RBCs were lysed with ACK buffer. All cells were transferred to a 96 U-shape well plate and kept on ice for the staining. Cells were stained as follows: first with Fc block CD16/CD32 (FRC/4G8) (BD Bioscience, 553142) for 15 min at RT in the dark. Cells were centrifuged and washed with FACS buffer and next stained with anti-mouse: CD8α clone 53-6, CD4 clone RM4-5 (BioLegend), CD19 clone 1D3 (BD Biosciences), CD11b clone M/170 (BioLegend), Ly6G clone 1A8 (BioLegend), Ly6C clone HK1.4 (BioLegend), PD-1 clone 29F.1A12 (BioLegend), and TIM-3 clone RMT3-23 (BioLegend) markers were used as exhaustion markers for CD8^+^ T cells. Sample analysis was performed using a BD LSRII and evaluated with Flojo (V.10) and GraphPad Prism (V10). The gating strategy involved a sequential selection of singlets, live/dead followed by lymphocytes or monocytes, and further categorized into CD8^+^, CD4^+^ T cells, CD19^+^ B cells, CD11b^+^ Ly6C^+^, and CD11b^+^ Ly6G^+^ MDSCs. 

### 5.13. High Dimensional Analysis of Flow Cytometry Data

High dimensional analysis of flow cytometry data was performed using t-distributed stochastic neighbor embedding (tSNE) and FlowSOM clustering and a visualizing technique. FCS files of TILs were used following compensation and gating on single cells and live cells using FlowJo X version 10.4.2. FCS files (4 FCS files from each group) were exported and then concatenated; each used file included a consistent number of events (180 events), and the total number of concatenated events was 720. The following markers were used: CD8, PD-1, and TIM-3. In FlowSOM analysis, five clusters were applied, as indicated in the figure legends. Histogram count was used to confirm marker expression. 

### 5.14. Sentinel Lymph Node Excision (SLNE)

Under isoflurane anesthesia (Attane, 800-544-7521), 10^5^ 4T1 cells were s.c. inoculated in the lower left quadrant of wild-type BALB/cOlaHsd mice. Mice were followed every 2 days to assess tumor growth and general health score. Eleven days later, the primary tumor was surgically dissected as follows: the weight of the mice was recorded, and 50 μL/25 g of narcotic mixture was injected s.c. Ophthalmic ointment (Paralube^®^) was applied to each mouse’s eyes to avoid the drying of the cornea. Ten minutes later, the mice were assessed for reflex responses, confirming their state of full sedation. Mice were positioned on their backs to expose the iLN, the skin of this area was shaved using a trimmer, and disinfection was carried out with Betadine. The hip joint was bent to an approximately 90° angle and a small incision of <5 mm through the skin was made using sterile curved micro-dissecting forceps and a surgical scissor. The incision was widened using the tip of the surgical scissor which enlarged the incision to about 10 mm. The iLN was localized and completely excised. The incision was sealed using sterile 9 mm wound clips inserted into a 9 mm auto-clip applier. Clips were removed after 10–12 days. Mice were monitored closely as per our license BE43/2021. 

### 5.15. In Vivo Cell Depletion Studies

For the depletion of CD8^+^, CD4^+^, and CD19^+^ cells, 300 μg of anti-CD8mAb clone CD8β Lyt3.2 (BioXcell, BE0223), anti-CD4mAb clone GK1.5 (eBioscience, 14-0041-82), and anti-CD19 mAb clone 1D3 (BioXcell, BE0150) was administered intravenously (i.v) on day 9 and 200 µg on day 21. Depletion efficiency was ~99% as determined by flow cytometry on day 14. 

### 5.16. Statistical Analysis

The data were analyzed and presented as mean ± SEM using GraphPad (V.8.4.2, Boston, MA, USA, www.graphpad.com) (464). Statistical comparisons among more than two groups employed one-way analysis of variance (ANOVA), while comparisons between two groups utilized the Student’s *t*-test. The survival rate was analyzed by log-rank (Mantel-Cox) test. Exact *p*-values were provided, and significance levels were denoted as **** *p* < 0.0001, *** *p* < 0.001, ** *p* < 0.01, and * *p* < 0.05.

## Figures and Tables

**Figure 1 vaccines-12-00355-f001:**
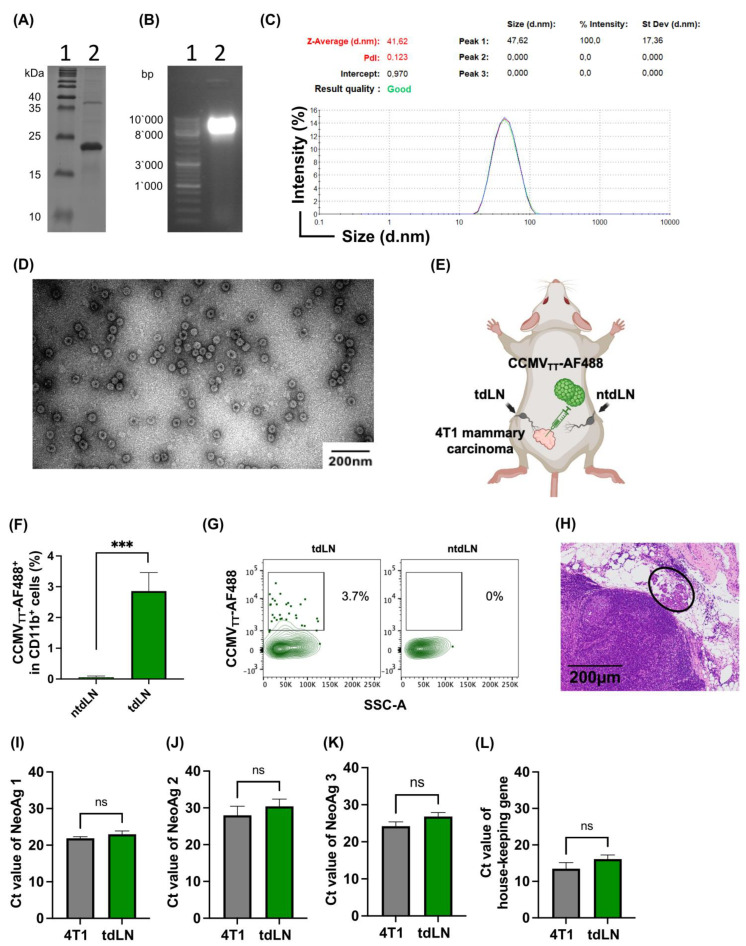
(**A**) SDS-PAGE showing a monomeric protein band of CCMV_TT_-VLPs at 25 kDa. (**B**) A total of 1% agarose gel stained with SybeSafe, Lane 1: 1 kB DNA marker, Lane 2: CCMV_TT_-VLPs spontaneously packaging ssRNA during the expression in *E*. *coli*. (**C**) Dynamic light scattering (DLS) of the VLPs. (**D**) The integrity of the particles was confirmed with electron microscopy, with images showing purified CCMV_TT_-VLPs, ~45 nm in size, next to a 200 nm scale. (**E**) Illustration of the anatomical localization of 4T1 tumor cells to ensure effective metastasis to the tdLN (iLN). (**F**) % CD11b^+^ cells uptake of CCMV_TT_-AF488 in tdLN and non-tdLN. The tumor-draining lymph node was used as a positive control and the negative control was the popliteal lymph node (beneath the tdLN). (**G**) Representative FACS plots of the CCMV_TT_-AF488 in the CD11b population within the tdLN and ntdLN. (**H**) HE staining of the tdLN, the black circle highlights a focal aggregate of metastatic epithelial tumor cells. (**I**–**L**) qrtPCR (Ct values) of 4T1 specific neoantigens in the tdLN, 4T1 cell-line was used as a +ve control and house-keeping gene Gapdh in (**L**). Statistical analysis (mean ± SEM) was conducted by the Student’s *t*-test in F. The sample size was n = 3. Significance levels are denoted as follows: *** *p* < 0.001 and ns = not significant.

**Figure 2 vaccines-12-00355-f002:**
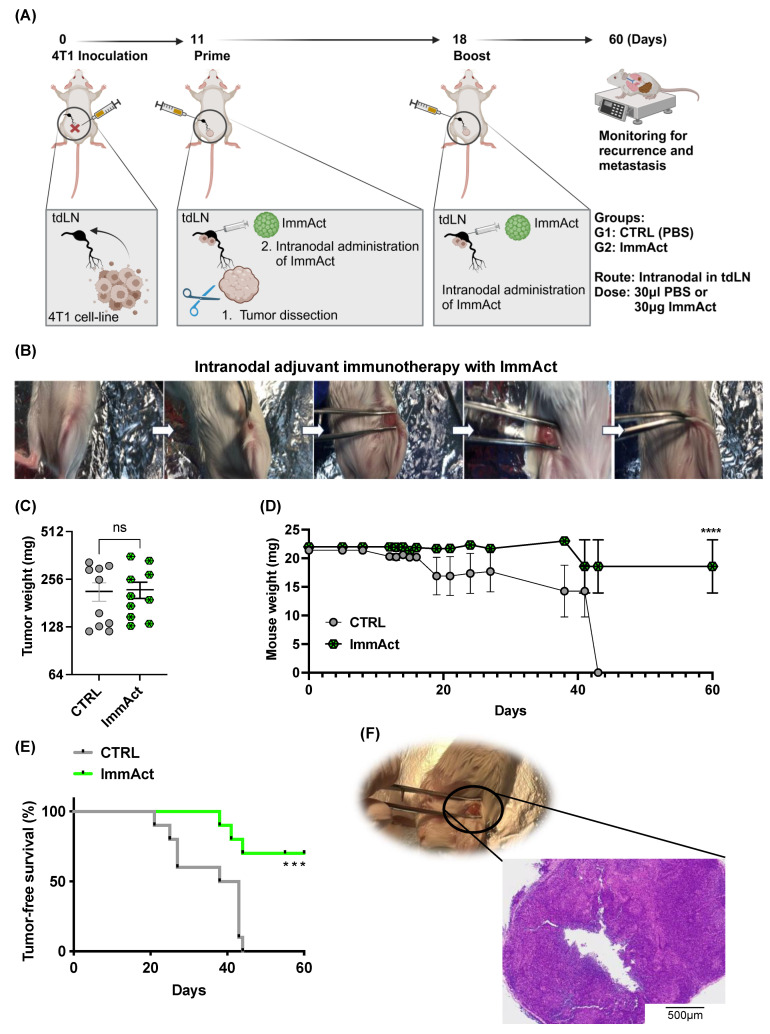
(**A**) Ilustration of the procedure, regimen, and the included groups of the adjuvant intranodal immunotherapy with ImmAct. (**B**) Images of the intranodal injection in mice (details in the method section). (**C**) Weight in (mg) of the primary tumors dissected on day 11 before the intranodal injection in the CTRL and ImmAct-treated group. (**D**) Mice weight in mg. (**E**) Percentage of tumor-free survival in both groups. (**F**) HE staining of the tdLN after intranodal treatment, scale bar at 500 μm. Statistical analysis by Student’s *t*-test and Log-rank test. The sample size was n = 10. Significance levels are denoted as follows: **** *p* < 0.0001; *** *p* < 0.001 and ns = not significant.

**Figure 3 vaccines-12-00355-f003:**
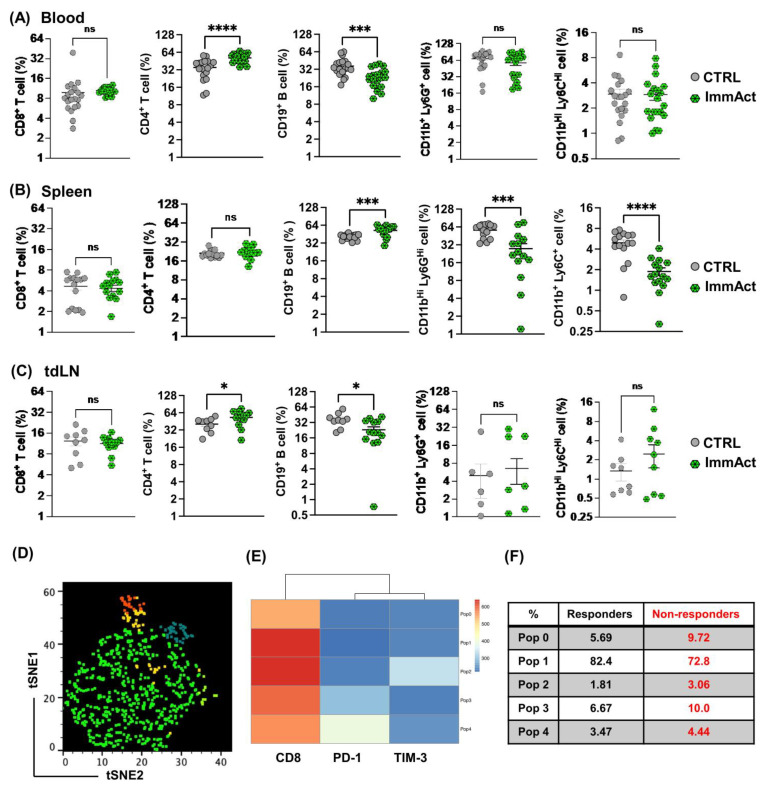
(**A**–**C**) Percentage of CD8^+^, CD4^+^ T cells, CD19^+^ B cells, CD11b^+^Ly6G^+^, and CD11b^+^ Ly6C^+^ cells in the blood, spleen, and the tdLN of mice in both groups. (**D**–**F**) High-throughput analysis of ImmAct immunized cohort, distinguishing responders and non-responders. (**D**) Concatenated tSNE clusters for CD8^+^ subsets in the responders cohort, revealing five distinct clusters highlighted by unique colors for enhanced visualization. Markers include PD-1 and TIM-3. (**E**) A heatmap presenting the expression of markers within each cluster. (**F**) A table comparing the percentage of each cluster in the responder and partial-responder cohorts. Statistical analysis (mean ± SEM) was conducted by Student’s *t*-test. The sample size was n = 20 in (**A**), n = 15 in (**B**), and n = 9 and 13 in (**C**). Significance levels are denoted as follows: **** *p* < 0.0001; *** *p* < 0.001; * *p* < 0.05 and ns = not significant.

**Figure 4 vaccines-12-00355-f004:**
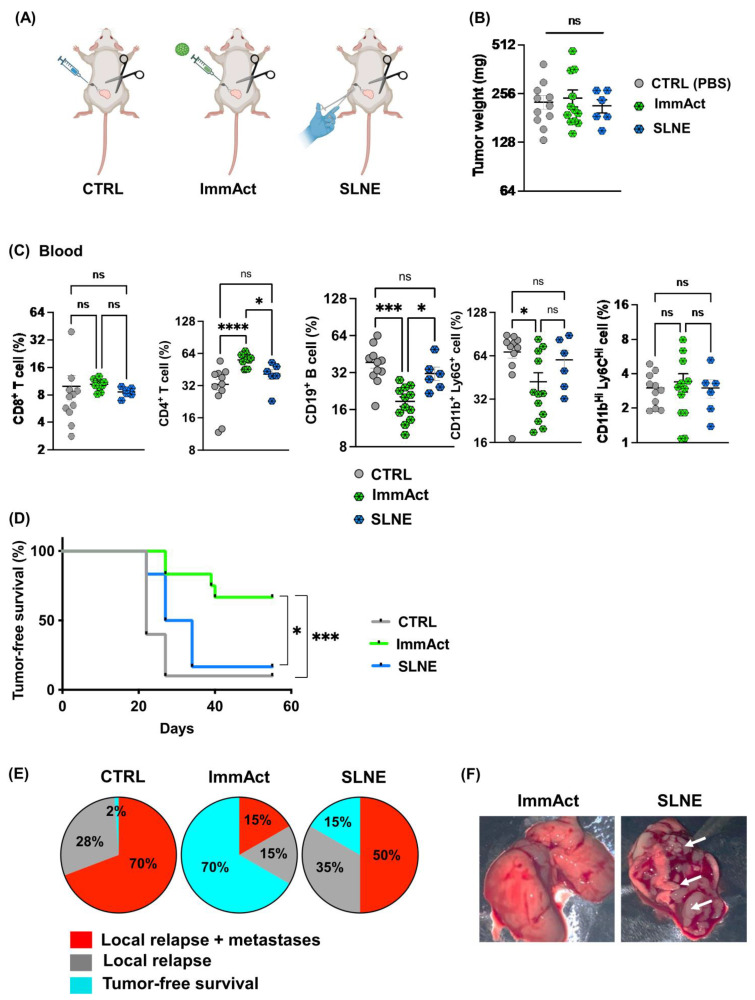
(**A**) Illustration of the three different groups: CTRL (PBS), ImmAct, and SLNE. The primary tumor dissection was performed on day 11 followed by SLNE and intranodal injections on days 11 and 18, respectively. (**B**) Weight (mg) of the primary tumors dissected on day 11. (**C**) Percentage of CD8^+^, CD4^+^ T cells, CD19^+^ B cells, CD11b^+^Ly6G^+^, and CD11b^+^ Ly6C^+^ cells in the blood of mice in the three different groups: CTRL, ImmAct, and SLNE. (**D**) Percentage of tumor-free survival in the different groups. (**E**) Pie chart demonstrating the percentages of tumor-free survival, local relapse, and local relapse with metastases in the three groups. Local relapse refers here to regrowth of tumor at the same site of the original location where the tumor cells had been inoculated. Metastatic disease was confirmed through lung tissue examination during necropsy, revealing the presence of metastatic nodules. (**F**) Representative images of the lung showing metastatic nodules in the ImmAct-treated group and SLNE group, highlighted by the white arrows. Statistical analysis (mean ± SEM) was conducted by one-way ANOVA and Log-rank test. The sample size was n = 6 or 12. Significance levels are denoted as follows: **** *p* < 0.0001; *** *p* < 0.001 and * *p* < 0.05; ns = not significant.

**Figure 5 vaccines-12-00355-f005:**
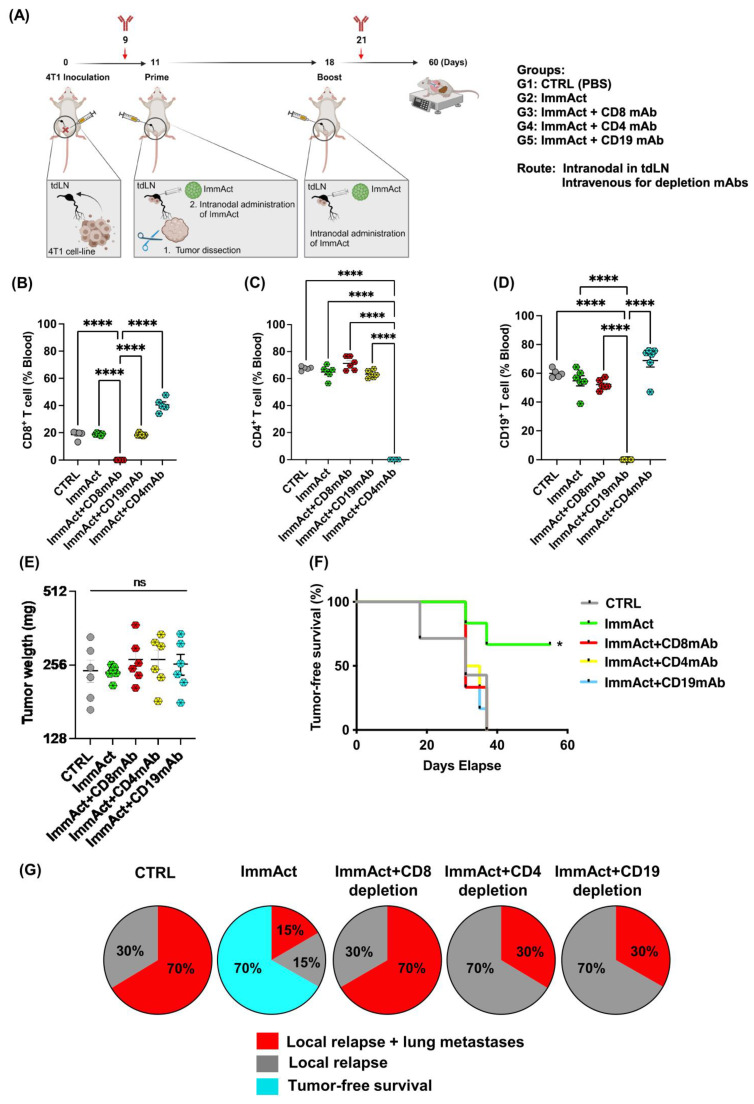
(**A**) Illustration of the experimental setup of immune cell depletion. The depletion antibodies were administered on days 9 and 21. The primary tumor was dissected on day 11, followed by intranodal injections on days 11 and 18. (**B**–**D**) Depletion efficiency, showing the percentage of CD8^+^ and CD4^+^ T cells and CD19^+^ B cells in the blood of the designed groups: CTRL, intranodal immunotherapy with ImmAct and intranodal immunotherapy with ImmAct combined with CD8, CD4, or B-cell depletion. (**E**) Weight in (mg) of the primary tumors dissected on day 11 before the intranodal injection in the different groups. (**F**) Percentage of tumor-free survival in each group. (**G**) Pie chart demonstrating the percentages of tumor-free survival, local relapse, and local relapse with metastases in the three groups. Local relapse refers here to the regrowth of the tumor at the same site of the original location where the tumor cells had been inoculated. Metastatic disease was confirmed through lung tissue examination during necropsy, revealing presence of metastatic nodules. Statistical analysis (mean ± SEM) was conducted by Student’s *t*-test and Log-rank test. The sample size was n = 6. Significance levels are denoted as follows: **** *p* < 0.0001 and * *p* < 0.05.

## Data Availability

Data are available upon reasonable request.

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
