# Peer review of "Intranodal Injection of Immune Activator Demonstrates Antitumor Efficacy in an Adjuvant Approach"

_vaccines, 2024, doi:10.3390/vaccines12040355_

Round 1

Reviewer 1 Report

Comments and Suggestions for Authors

The study presents a novel approach to cancer immunotherapy, focusing on the transformation of tumor-draining lymph nodes (tdLN) from an immunologically "cold" to a "hot" site using a universal immune activator (ImmAct). Through rigorous methodologies, including in vivo and in vitro assays and flow cytometry, the research demonstrates that intranodal ImmAct administration post-primary tumor dissection improves tumor-free survival and minimizes weight loss in a murine model. The findings highlight not only the local and systemic immunological alterations induced by ImmAct but also reveal variability in immune responses among individual subjects. The study is comprehensive and well-written, and I have a few comments that I would like you to consider.

1.     In relation to figure 1c, it would be preferable to redraw it by utilizing the text data. Additionally, it is suggested that you include a statistical analysis of your particle's size, both for DLS data and TEM data, as this would be beneficial for comparison purposes.

2.     It is worth noting that at 40 days, there was a decrease in weight for both CTRL and ImmAct groups. Could you provide more detailed explanations in your manuscript?

3.     The study claims that ImmAct outperforms the current gold standard of sentinel lymph node excision, it would benefit from deeper mechanistic insights and comparisons with existing treatments to enhance its clinical relevance. 

Reviewer 2 Report

Comments and Suggestions for Authors

Comments:

1. In humans, the lymph nodes which are invaded by cancer cells were usually removed by surgery. At this report, authors claim the intranodal injection of immune activator could be used to replace nodal removal by surgery. Would authors explain in details and discuss this topic more in discussion.

2. The current study uses mammary carcinoma as an example. Please explain why. Any similar data from other types of cancers?

3. Please provide the protein sequence of immune activator.

4. Do authors measure ILs in this study? If yes, please provide the data.

5. This mice study covered the period of 60 days. The average life span of mice is 3 years. Would this be comparable to humans in terms of effectiveness? Also the metabolic rate in mice is much higher than that in humans. Would this be a limitation to apply to humans? Please discuss.

Reviewer 3 Report

Comments and Suggestions for Authors

Romano Josi and colleagues present a high quality and well-written experimental manuscript focused on intranodal injection of immune activator demonstrates antitumor efficacy in an adjuvant approach.

Authors explored the efficacy of a universal immune activator (ImmAct) targeted to the tdLN. This approach can be viewed as an attempt to turn a cold unresponsive tdLN, into a hot, responsive site. The adjuvant antitumor efficacy of novel intranodal injection was evaluated in a metastatic mammary carcinoma murine model. The cancer cells were inoculated subcutaneously in the lower quadrant of the mouse to provoke the tdLN (inguinal lymph node). The study encompasses a range of methodologies, including in vivo and in vitro assays and high-dimensional flow cytometry analysis. 

Authors utilized their immunogenic plant-virus-like nanoparticles derived from cowpea chlorotic mosaic VLPs (CCMVTT-VLPs referred to as ImmAct). They have produced and fully characterized the VLPs. The 45nm icosahedral nanoparticles package ssRNA -TLR7/8 ligand- during the expression process in the E. coli host system. CCMVTT-VLPs also incorporate an internal tetanus-toxin epitope, known to activate T helper cells. They then worked on establishing a murine model in which tumor cells from the established primary tumor are capable of metastasizing to the tdLNs. They used the aggressive 4T1 murine mammary carcinoma cells which have the potential to form lymphangiogenesis and metastatic disease. 

Their findings demonstrated that intranodal administration of ImmAct following the dissection of the primary tumor led to improved tumor-free survival and minimized weight loss. ImmAct led to both local and stemic alterations in the cellular and humoral immunity. Additionally, after ImmAct treatment, non-responders showed a higher rate of exhausted CD8+ T cells compared to responders. Indeed, their innovative approach surpassed the gold standard surgery of sentinel lymph node excision.

Finally, authors conclude that their study successfully highlights the effectiveness of an intranodal immunotherapy employing innovative immune activator derived from plant nanoparticles, loaded with potent immune stimuli. The administration of ImmAct in an adjuvant setting, (i.e. post-primary tumor dissection), involving only a prime and a boost injection, not only highlights the robust efficacy of immunotherapy but also paves the way for future optimization in preparation for human translation.

Overall, the manuscript is highly valuable for the scientific community and should be accepted for publication.

======================

Other comments to authors:

1) Please check for typos throughout the manuscript.

2) Please improve figures/tables where appropriate. 

3) Lines 308-320. With regards to T cell exhaustion – authors are kindly encouraged to cite the following article that describes a significant role of various factors in T-cell quiescence, differentiation and exhaustion.
DOI: 10.3389/fimmu.2022.971045

Reviewer 4 Report

Comments and Suggestions for Authors

The current manuscript is an interesting experimental study on the intranodal injection of an immune activator as adjuvant anticancer therapy. It appears to be overall well-done, with many relevant assays having been performed. Nevertheless, some quite relevant alterations are necessary before acceptance for publication:

- The iThenticate report shows a high degree of similarity with existing sources, the text should be rewritten in the identified places before the manuscript can be considered for acceptance; this is a major flaw, the authors should correct this ASAP;

- In the introduction section, more should be said on current cancer therapies and their limitations;

- The introduction section should end with the study’s objective, and not with the main results and conclusions, this should be saved for the respective sections;

- All sections and subsections should be adequately numbered;

- The sought approach includes an invasive administration method; could this type of therapy work through non-invasive administration routes? If so, which one(s), and which formulation type and composition should be selected?;

- In the discussion section, this study’s results should be compared to other previous studies using similar methods; the appropriate references should also be added;

- The method “CCMVTT-VLPs cloning, expression, and production” should be adequately described, authors have just inserted the reference, and this forces readers to have to check it; at least a minimum method description should be provided, along with the respective reference;

- Some details are missing from DLS method description, such as cuvette type and composition and sample dilution;

- Nanoparticle size and PDI should be reported;

- Cell culture methods should be much more detailed;

- A section on statistical analysis should be added;

- An abbreviation list should be added.

Round 2

Reviewer 2 Report

Comments and Suggestions for Authors

No more comments.

Reviewer 4 Report

Comments and Suggestions for Authors

The authors have made most necessary changes.